# LLM-FMS: A fine-grained dataset for functional movement screen action quality assessment

**Qingjun Xing**[1], **Xuyang Xing**[2], **Ping Guo**[3], **Zhenhui Tang**[4], **Yanfei Shen**[5]*

1 School of Sport Science, Beijing Sport University, Beijing, China, 2 Department of Automation, Najing University of Science and Technology, Najing, Jiangsu, China, 3 Intel Labs China, Beijing, China, 4 Department of Automation, Shanghai Jiao Tong University, Shanghai, China, 5 School of Sport Engineering, Beijing Sport University, Beijing, China

* syf@bsu.edu.cn

**Data availability statement:** This work presents a fine-grained dataset of FMS action

## Abstract

The Functional Movement Screen (FMS) is a critical tool for assessing an individual's basic motor abilities, aiming to prevent sports injuries. However, current automated FMS evaluation is based on deep learning methods, and the evaluation of actions is limited to rank scoring, which lacks fine-grained feedback suggestions and has poor interpretability. This limitation prevents the effective application of automated FMS evaluation for injury prevention and rehabilitation. We develop a fine-grained, hierarchical FMS dataset, LLM-FMS, derived from FMS videos and enriched with detailed, hierarchical action annotations. This dataset comprises 1812 action keyframe images from 45 subjects, encompassing 15 action representations of seven FMS actions. Each action includes a score, scoring criteria, and weight data for body parts. To our extensive knowledge, LLM-FMS is the first fine-grained fitness action dataset for action evaluation task. Additionally, a novel framework for action quality assessment based on large language models (LLMs) is proposed, designed to enhance the interpretability of FMS evaluations. Our method integrates expert rules, utilizes RTMPose to extract key skeletal-level action features from key frames, and inputs prompts into the LLM, enabling it to infer scores and provide detailed rationales. Experimental results demonstrate that our approach significantly outperforms existing methods while offering superior interpretability. Experimental results demonstrate that our approach outperforms existing methods in terms of accuracy and interpretability, with a substantial increase in the clarity and detail of the rationales provided. These findings highlight the potential of our framework for fine-grained action quality assessment with the aid of LLMs.

## Introduction

In contemporary lifestyles, poor behavioral habits and lifestyle negatively affect individual physical function. This impact is particularly evident during physical activity, manifesting primarily as reduced exercise capacity [1,2]. The Functional Movement Screen (FMS) is a widely used, efficient tool in sports medicine for screening fundamental movement patterns.

keyframes, which can be accessed via the following Github repository: https://doi.org/10.6084/m9.figshare.c.7601630.v1.

**Funding:** This study has been supported by the National Natural Science Foundation of China, Grant No. 72071018, the Fundamental Research Funds for the Central Universities of China, No. 2023GCZX003 (Research on Nonlinear Accurate Measurement System of Exercise Loads).

**Competing interests:** The authors have declared that no competing interests exist.

It quickly identifies deficiencies in these patterns, effectively reducing the risk of sports injuries [3,4]. FMS consists of seven fundamental movement patterns: deep squat, hurdle step, inline lunge, shoulder mobility, active straight leg raise, trunk stability push-up, and rotational stability. These movements assess flexibility, stability, and motor control. To assess symmetry and balance, movements are performed on both sides, totaling 15 test movements, as illustrated in Fig 1. The combination of these movements provides a comprehensive perspective for assessing an individual's physical functionality.

The effectiveness of the FMS lies in its scoring system, which quantifies the quality and stability of individual movements [3]. The scoring system consists of four levels: 0, 1, 2, and 3 points. Higher scores indicate better movement proficiency and stability. It is important to note that if the subject experiences pain during the assessment, the movement is scored as 0. The remaining three levels reflect the degree of movement execution and stability, provided there is no discomfort or pain. FMS has been shown to aid in injury prevention and has broad applications in sports training, rehabilitation, and public fitness [4–6].

Traditional FMS assessment methods rely on experts' on-site visual inspection and palpation. However, these methods have several limitations. First, the assessment process is time-consuming, and the scoring results may be influenced by the subjective experience of the evaluators. Second, there is a shortage of experts with extensive experience, and their distribution is geographically imbalanced, particularly in economically underdeveloped regions where access to professional assessment resources is limited [5,7,8]. Therefore, utilizing computer technology and artificial intelligence to automate and enhance functional movement screening is crucial for improving assessment efficiency, minimizing subjective bias, and broadening service coverage [9].

Recent studies have confirmed the feasibility of FMS objective evaluation automation. In 2014, Whiteside et al. [9] compared FMS scores assigned by human testers to those generated by a motion capture system. Their results demonstrated that, compared to manual scoring,

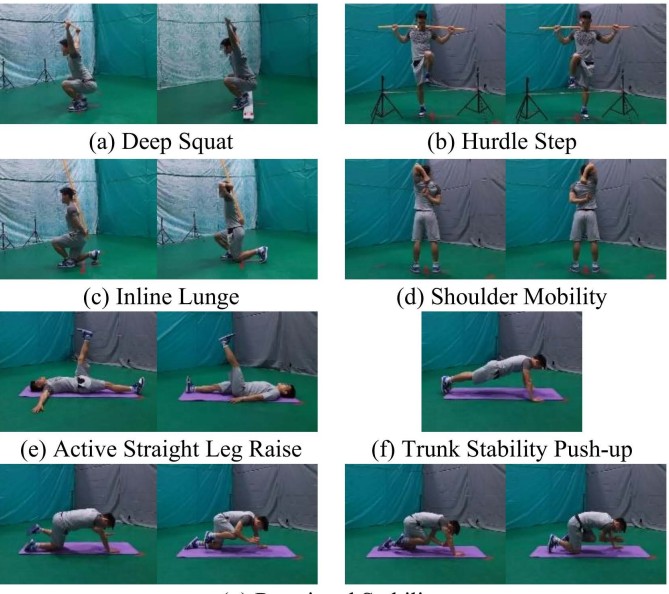

(a) Deep Squat  (b) Hurdle Step

(c) Inline Lunge  (d) Shoulder Mobility

(e) Active Straight Leg Raise  (f) Trunk Stability Push-up

(g) Rotational Stability

**Fig 1. The 15 movements of FMS.** From left to right, from top to bottom.

the automated FMS scoring system provided more objective assessments, significantly reducing the influence of subjectivity. Hong et al. [10] proposed an automated FMS assessment method based on an improved Gaussian mixture model. This method extracted action features corresponding to different scores and trained a Gaussian mixture model, utilizing maximum likelihood estimation for scoring. The results showed high consistency with expert evaluations. Unlike the aforementioned methods that rely on manual feature extraction, Lin et al. [11] introduced an automatic FMS assessment method using deep learning. This approach employs the I3D network for video feature extraction, incorporating an attention mechanism and a multi-layer perceptron to capture spatio-temporal video features at multiple scales and levels, thereby improving the accuracy and reliability of the evaluation. Shen et al. [12] developed an FMS assessment framework based on a multi-view deep neural network (MVDNN). This framework combines automatic skeleton feature extraction with manual feature selection, extracting three-dimensional trajectory features of movements from two perspectives. By integrating automatic video analysis with deep learning, this framework can assess the quality of an individual's FMS performance without the need for physical markers. Also in order to realize FMS automatic evaluation, Lin et al. [13] proposed an automatic assessment method for functional motion screening based on two-stream network and feature fusion. The proposed method uses RAFT algorithm to estimate the video optical flow, which can better capture the spatio-temporal characteristics compared with the single-flow method. At the same time, the method uses attention fusion to combine optical flow features with the original frame, which improves the prediction accuracy. The proposed framework outperforms the method in [11].

While the automated assessment methods ensure objective action scoring and effectively reduce the influence of expert subjectivity, there are still several problems. The datasets used in these studies contained only expert scores and lacked fine-grained annotations, limiting the ability of assessment methods to provide detailed scoring feedback. In addition, most deep learning-based models are black-box systems, rendering the decision-making process opaque. Consequently, users are unable to receive actionable feedback or recommendations, hindering the potential for subsequent training and improvement.

Large language models (LLMs) are a class of artificial intelligence models with powerful contextual understanding and reasoning capabilities. Recent studies have demonstrated the potential of LLMs in action analysis and motor control. Zhao et al. [14] leverage the contextual modeling and reasoning capabilities of LLMs, as well as the potential of multi-modal fusion, to propose a two-stage framework called AntGPT. It first identifies actions that have been performed in the observed video text data, and then asks the LLM to predict future actions through conditional generation, or infer goals and plan the entire process through thought prompts. This framework achieves the best performance on benchmarks such as Ego4D LTA and EPIC-Kitchens-55. Joublin et al. [15] proposed a hierarchical replanning architecture, which exploits the commonsense knowledge and implicit reasoning capabilities of LLMs to implement corrective replanning strategies in robot task planning, and verified the effectiveness of the architecture in simulated and real world environments. The study demonstrates the potential of LLMs to deal with physically grounded, logical, and semantic errors, and how feedback can be used to reevaluate and adjust plans in a timely manner.

We believe that LLMs are able to facilitate a fine-grained level of action quality assessment (AQA), and therefore constructed a new FMS evaluation dataset, LLM-FMS, which is the first fine-grained dataset to utilize LLMs for assessing action quality. LLM-FMS has two key features: (1) A three-level semantic structure. All keyframes of FMS are annotated with three levels of semantic tags: scores, scoring details, and body part information. The

scoring details provide specific explanations of the action performance, enabling the model to better interpret action quality; (2) The dataset contains action keyframes and corresponding fine-grained annotation files, and its format is optimized for generating prompts for large language models.

This paper also proposes a framework that leverages the reasoning capabilities of LLMs to evaluate action quality on the LLM-FMS dataset (Fig 4). The framework first utilizes the open-source pose estimation tool, RTMPose [16], to extract skeletal data from FMS keyframes, followed by the extraction of action feature evaluation metrics based on predefined scoring rules. These evaluation results are then embedded into a prompt and fed into the LLM, which assigns scores to the FMS actions and provides detailed scoring feedback according to expert rules embedded in the prompt. To the best of our knowledge, this is the first method to apply LLM for the fine-grained automated assessment of FMS. In summary, the main contributions of this paper are as follows:

- We constructed the first fine-grained dataset for FMS. Building on the publicly available FMS dataset by Xing et al. [17], we extracted keyframes from each action segment and performed a three-level semantic annotation, including action scores, scoring details, and body parts associated with scoring points, as illustrated in Table 1. This dataset represents the first fine-grained scoring resource for FMS action assessment.

- We developed a model with strong interpretability based on LLMs. Unlike traditional black-box models, our method offers greater transparency, as the entire assessment process is grounded in well-defined expert rules and knowledge bases. This allows users to easily comprehend the model's decision-making process. The enhanced transparency not only improves the model's credibility but also increases its acceptability and potential for adoption in practical applications.

- Fine-grained action assessment and improvement feedback. Our method not only provides a score but also accurately identifies specific action errors and offers targeted improvement suggestions based on expert knowledge. This approach delivers practical, actionable feedback to testers and athletes, enabling them to refine specific techniques and enhance overall sports performance.

## Methods

### LLM-FMS fine-grained dataset

In this section, we introduce a novel fine-grained FMS action keyframe dataset, LLM-FMS. We will present details on its construction and statistical properties.

**Dataset construction.** We extract RGB image data from the FMS multimodal dataset publicly available by Xing et al. [17]. Utilizing the RTMPose, we extracted the skeletal

**Table 1. Three-layer semantic structure of the dataset.**

| Action | Score | Scoring details | Body parts |
|---|---|---|---|
| Deep squat[a] | 3 | Angle 1: Trunk-calf angle. The trunk is parallel to the shins. | Trunk, left lower limb, right lower limb |
| | | Position 1: Hip height. Hip lower than knee. | left lower limb, right lower limb |
| | | Position 2: Wrist position. The wrist joint is to the right of the knee joint. | Left upper limb, right upper limb, left lower limb, right lower limb |

[a]Taking deep squat as an example.

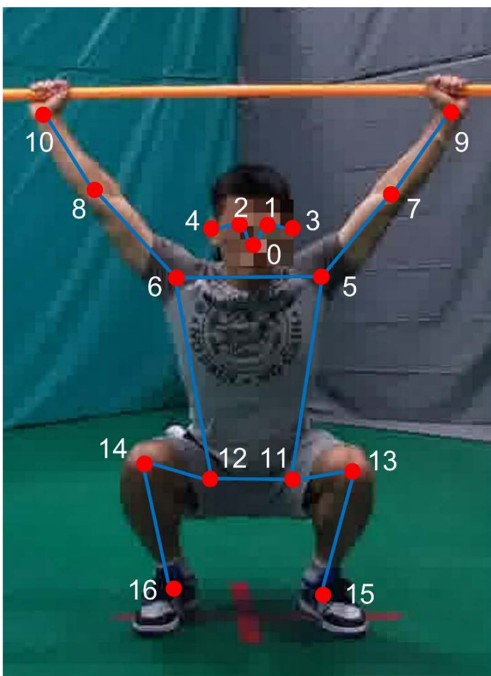

| | |
|---|---|
| 0 | Nose |
| 1 | Left eye |
| 2 | Right eye |
| 3 | Left ear |
| 4 | Right ear |
| 5 | Left shoulder |
| 6 | Right shoulde |
| 7 | Left elbow |
| 8 | Right elbow |
| 9 | Left wrist |
| 10 | Right wrist |
| 11 | Left hip |
| 12 | Right hip |
| 13 | Left knee |
| 14 | Right knee |
| 15 | Left ankle |
| 16 | Right ankle |

**Fig 2. Human skeleton.**

data for each frame, as illustrated in Fig 2, and calculated the cosine similarity according to rules to identify keyframe images from each action sequence. Finally, we enlisted experienced FMS experts to re-score the keyframe images and perform fine-grained annotations.

**Dataset dictionary.** We constructed a fine-grained keyframe dataset characterized by a three-level semantic hierarchy, which includes action scores, scoring details, and body part information, as illustrated in Table 1. To facilitate this process, we engaged an experienced, FMS-certified expert to assist in establishing the rules and performing the three-level semantic annotation of the action keyframe images.

Regarding the semantic structure illustrated in Table 1, the scoring label denotes the specific score assigned to an individual's action, while the scoring detail label provides specific information regarding the rationale behind that score. The scoring details vary across different actions. Additionally, the body part label identifies the primary body parts involved in the action, with the quality of limb movement in these areas directly influencing the score.

This study performs a fine-grained assessment of FMS actions utilizing the above three-level semantic hierarchy.

**Dataset annotation.** We have developed a desktop annotation tool, illustrated in Fig 3, to enable FMS experts to conduct fine-grained semantic annotations of action keyframes more efficiently. Given a keyframe image, experts annotate each action using a predetermined dictionary. The annotation process consists of two stages, progressing from coarse to fine granularity. The coarse-grained stage involves marking the score for each action instance, while the fine-grained stage entails detailing the scoring aspects that contribute to the overall action score and recording the key body parts involved.

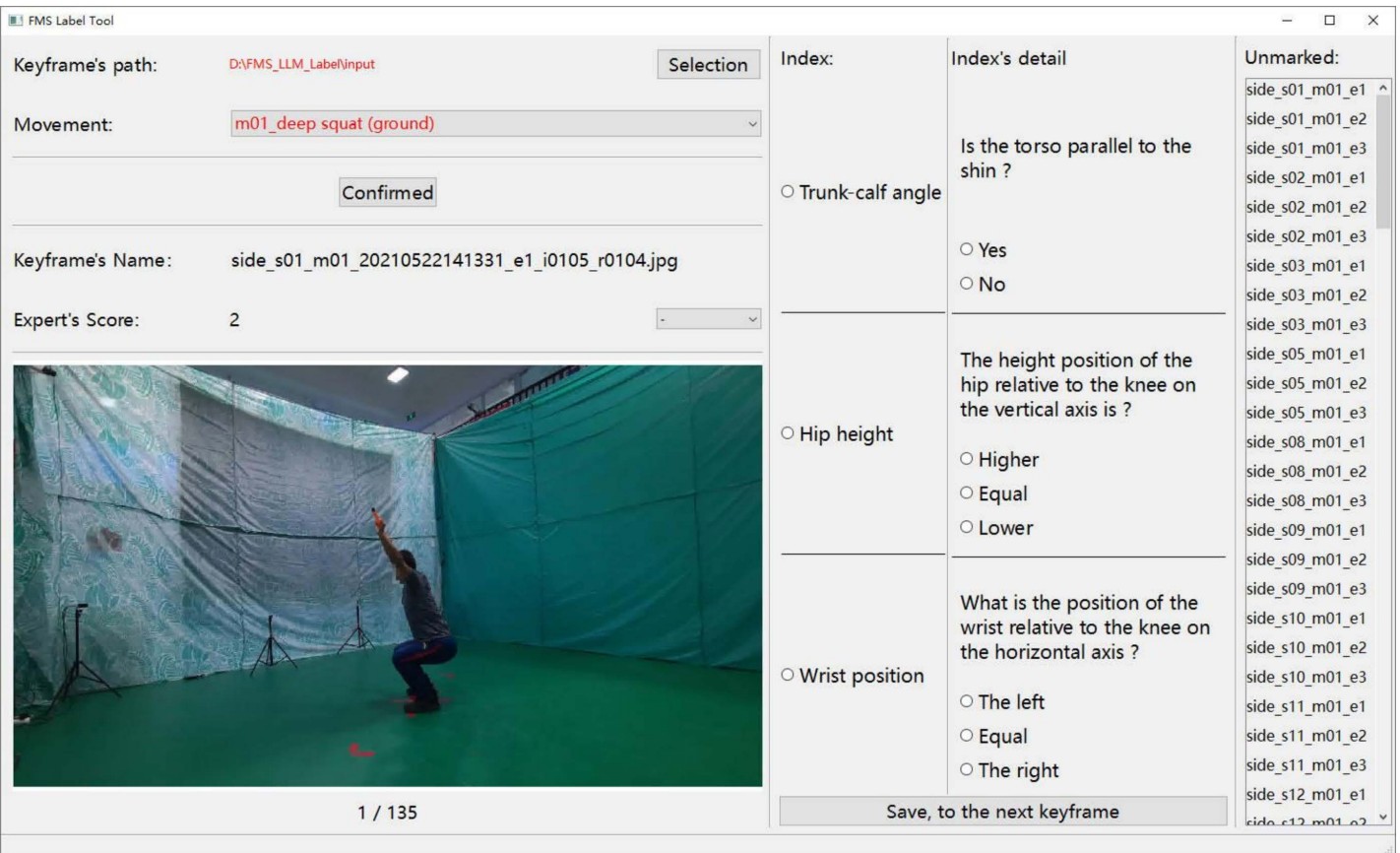

**Fig 3. Annotation tool.**

The entire annotation process is conducted by a single FMS expert to ensure consistency across all keyframe annotations. The total duration of the annotation process is approximately 40 hours.

**Basic information of the dataset.** The LLM-FMS dataset comprises 1,812 keyframe images from 45 subjects, encompassing 15 distinct difficulty levels FMS actions, as illustrated in Fig 1. The demographic information of the subjects is in S1 Table. All subjects have signed informed consent and agreed to share their experimental data for scientific research. This study was also reviewed and approved by the Ethics Committee of Sports Science Experiment of Beijing Sport University (Approval number: 2021156H). Each action is associated with a score, several scoring details, and body part information. These data facilitate a comprehensive assessment of the flexibility of the individual's left and right sides and allow for the examination of performance under varying difficulty levels. Table 2 presents detailed information about our dataset and compares it with existing AQA datasets for sports fitness and rehabilitation. Our dataset differs from existing AQA datasets in terms of granularity. For instance, datasets such as Fitness-28, UI-PRMD, and 3D-Yoga provide only action scores or ratings, whereas our dataset offers not only action scores but also fine-grained semantic annotations, including scoring details and body part information. Consequently, the absence of fine-grained semantic annotations in other datasets limits them to merely scoring or grading actions, thus precluding a comprehensive semantic-level action quality evaluation. To our knowledge, LLM-FMS is the first fine-grained dataset for LLM to evaluate the quality of fitness actions.

## LLM-based assessment framework

In this section, we systematically introduce the AQA framework based on LLMs, which facilitates fine-grained evaluations of users' actions and offers targeted improvement suggestions. The overall architecture of our method is illustrated in Fig 4.

**Problem definition.** Given an action image sequence $S$ and the corresponding scoring rules $R_s$ and $R_k$, the proposed framework is formulated as a classification problem, assessing

**Table 2. Comparison of existing rehabilitation and fitness datasets with LLM-FMS.**

| Dataset | #Samples | #Act. Clas. | Moda. | Anno.Type |
|---|---|---|---|---|
| Fitness-28 [18] | 7530 | 28 | V, S | Similarity |
| Uco Physical Rehabilitation [19] | 2160 | 8 | V, S | Score |
| KIMORE [20] | 1950 | 5 | R, S | Score |
| UI-PRMD [21] | 1100 | 10 | S | Grade |
| 3D-Yoga [22] | 3792 | 117 | V, S | Score |
| **LLM-FMS** | **1812** | **15** | **R, S** | **Score, Step** |

Note：V: Video modality; S: Skeleton modality; R: RGB modality; Similarity: similarity annotation; Score: score annotation; Grade: rating annotation; Step: Fine-grained semantic annotation.

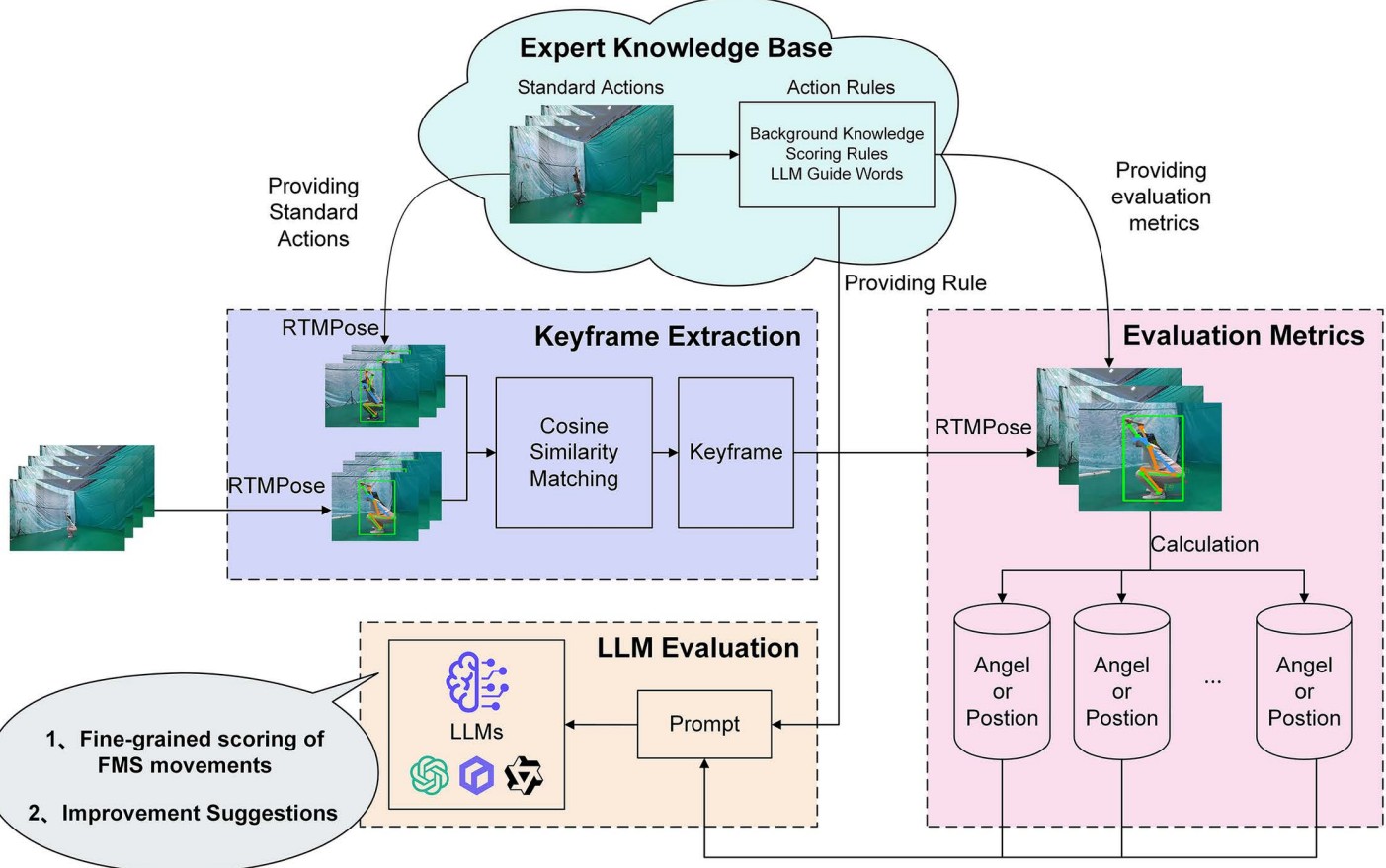

**Fig 4. LLM-based fine-grained FMS action evaluation process framework.**

actions using a LLM to generate fine-grained evaluations of action quality. This can be represented as follows:

$$k = \mathcal{C}\left(S|R_s\right) \qquad (1)$$

$$\hat{Y} = \mathcal{L}\left(\mathcal{F}\left(k|R_k\right)\right) + Y \qquad (2)$$

Here, $S = \{k_1, k_2, k_3, \cdots, k_n\}$ represents the action image sequence, $R_s$ represents the threshold discriminant condition required for action key frame extraction, $C$ represents the keypoint cosine similarity matching function, which is used to extract action key frames $k$ from the action image sequence. $R_k$ represents the scoring rules of keyframe actions, $F$ represents the scoring index extraction function of keyframe actions, and calculates the specific scoring index information of actions according to the scoring rules $R_k$. $L$ represents a large language model. By embedding the scoring index information extracted in the previous step into prompt, with the help of LLM's active logical reasoning ability, the score of the action $\hat{Y} = \{1, 2, 3\}$ is given. $Y = \{1, 2, 3\}$ is the truth score of the action keyframe.

**Construction of knowledge rule base.** Based on the FMS scoring rules [3] and consultations with FMS experts, the keyframes of the seven FMS movements and their corresponding scoring criteria have been incorporated into the FMS movement assessment knowledge base. The specific process is as follows:

(1)Action keyframe selection.

Typically, FMS assessment evaluates the whole action time series. However, in this study, we focus on action keyframes. Therefore, FMS experts were consulted to establish threshold conditions for the angles and distances of 15 movements, in line with traditional FMS scoring criteria, see S1 Text. Based on these threshold conditions, the selected keyframes for the 15 movements are illustrated in Fig 1. It is important to note that, to facilitate the calculation of angle and position information for subsequent visual tasks, side views were chosen for the deep squat, hurdle step, and inline lunge, while front views were used for the remaining movements.

(2)Action rule definition.

This study employs manually extracted features from FMS keyframes of skeletal key points to conduct a fine-grained assessment of movement quality. These features are designed based on domain-specific knowledge and expert experience. Despite advancements in automatic feature extraction methods using deep learning, manually extracted features remain valuable for analyzing human skeletal behavior.

FMS experts standardized the scoring criteria for each movement keyframe in accordance with FMS scoring guidelines. These criteria include joint position, angle, and distance features, which are used to grade movement quality. The definitions of these features are outlined as follows.

- **Position features** represent the relative positional relationships between joints.

- **Joint angle features** denote the angles formed either between adjacent joints or between a joint and a reference coordinate axis.

- **Distance features** describe the spatial distances between joints.

Additionally, FMS experts refined the identification of body parts critical to the execution of different movements. They then re-evaluated the scoring (ranging from 1 to 3 points) for each

movement keyframe based on the dataset by Xing et al. [17], as illustrated in Fig 3. For the deep squat, the experts clearly defined the standard angle range between the trunk and lower leg, as well as the required hip height, along with relevant movement criteria. Detailed rules for other movements are provided in S2 Text.

**Keyframe extraction.** This study focuses on extracting skeletal features from action keyframes for fine-grained movement assessment. Accurate identification of keyframes from image sequences is essential. First, the RTMPose is applied to extract the coordinates of 17 skeletal key points from each image in the action sequence, as shown in Fig 2. Next, the skeletal data are normalized, and the right hip joint point is selected as the origin of the skeleton coordinate system. The sum of the cosine angles between the vectors formed by other key points and the origin is calculated to represent the human posture. Finally, by comparing the Euclidean distance between the standard action frame and the sum of the cosine angles for each frame in the sequence, the action keyframe is identified. The detailed calculation process is as follows:

The right hip joint $\left(x_{rightHip}, y_{rightHip}\right)$ is selected from the human skeleton coordinates of 17 key points as the origin of the skeletal coordinate system. Subsequently, the sum of the cosine angles between the vectors formed by the other key points and the origin is calculated as follows:

$$\Theta_K = \sum_{i=0}^{K} \frac{\left(x_{rightHip}, y_{rightHip}\right) \cdot \left(x_i, y_i\right)}{\sqrt{x_{rightHip}^2 + y_{rightHip}^2} \sqrt{x_i^2 + y_i^2}}, K = 16 \tag{3}$$

Similarly, by calculating the sum of the cosine angles for the skeletal points in the standard action keyframe, denoted as $\Theta_N$, the Euclidean distance between an image and the standard action keyframe is represented as follows:

$$D = \left|\Theta_K - \Theta_N\right| \tag{4}$$

Finally, the image with the smallest value is selected as the action keyframe, as follows:

$$K = arg \quad min \ D \tag{5}$$

For all 15 movements, standard action keyframe images have been established. Using the cosine similarity calculation method described above, we identified 1,812 keyframes across the action sequences for these movements.

**Action scoring indicator calculation.** Based on the scoring indicators for each movement established in Section 4.2, we calculate the scoring indicators for the user's movements to facilitate subsequent assessments. RTMPose is employed to extract the key points from the keyframes and compute the corresponding evaluation metrics based on these indicators and the extracted key point coordinates, as illustrated in Fig 5.

For example, for the deep squat, it is essential to calculate the angle ($\theta_{5-11, 13-15}$) between the trunk and the lower leg, which is defined by the vectors formed by key points 5 and 11 and key points 13 and 15. Additionally, the positional relationship between the hip joint and the knee joint must be assessed. This relationship is determined by comparing the y-coordinates of key points 11 and 13; specifically, if the y-coordinate of key point 11 is less than that of key point 13, the hip joint is positioned lower than the knee joint; otherwise, the hip joint is positioned higher. Similar calculations apply to other movements, where angles are computed directly. For positional relationships, coding is required to evaluate key points and subsequently output the positional information, facilitating reasoning in subsequent large models.

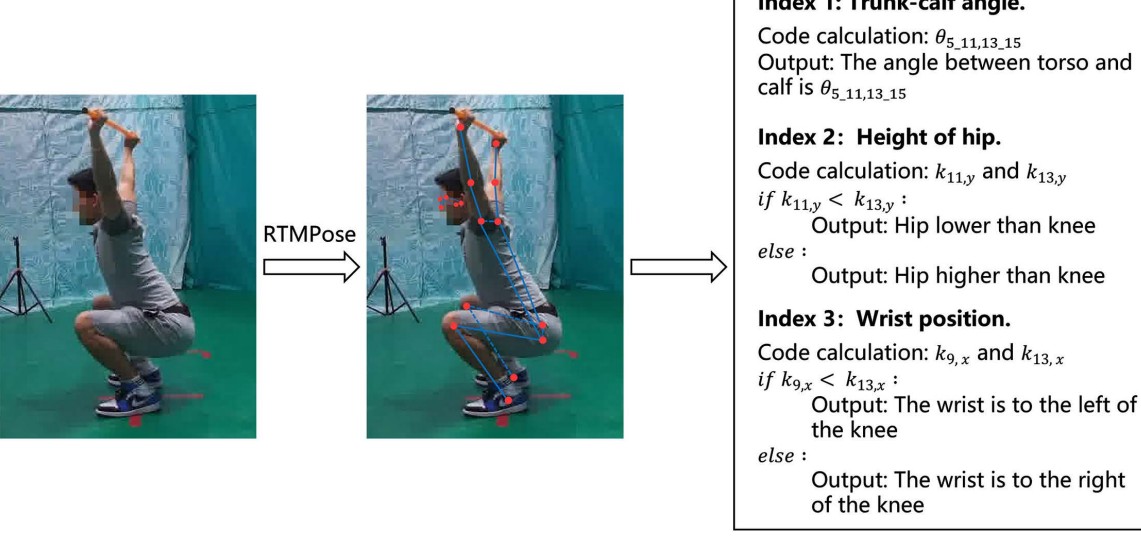

**Fig 5. Calculation of action score index.** Taking deep squat as an example.

**LLM prompt generation and AQA.** Following the processing steps outlined in Sections 4.2 to 4.4, to achieve fine-grained FMS using LLMs, it is also necessary to generate appropriate prompt for each movement. The design of prompts plays a critical role in the use of LLMs, as it directly influences the quality and relevance of the model's output.

In this study, for each movement, prompts are generated based on expert-defined scoring rules and the output of angle or positional information from the user's keyframes. The specific content of these prompts is provided in S3 Text. Once the prompt is generated, it is input into the LLM, which then produces a fine-grained score and offers improvement suggestions for the movement.

## Results

### Evaluation metrics

Building on previous work [10–13], we evaluate the model's performance on FMS movements using predicted score accuracy, macro-averaged F1 (maF1), and the Kappa index. Additionally, we assess the interpretability of the proposed framework for fine-grained movement quality assessment.

**Scoring accuracy.** Scoring accuracy quantifies the alignment between the predicted movement scores generated by the framework and the ground truth labels. In this study, scoring accuracy is calculated using the following formula:

$$Accuracy = \frac{N_{normal}}{N_{total}} \tag{6}$$

Where $N_{normal}$ is the number of samples for which the model prediction is correct, that is, the number of samples for which the model predicted action score is consistent with the expert rating. $N_{total}$ is the total number of samples of FMS, that is, the number of all samples participating in the evaluation.

**Macro-averaged F1 (maF1).** maF1 is employed to assess the accuracy of multi-class classification problems. In such scenarios, maF1 first computes the F1 score for each category

and subsequently averages these scores. Specifically, if there are $C$ categories, with each category's F1 score denoted as $F_1^1$, $F_1^2$, $\cdots, F_1^C$, the maF1 score is calculated as follows:

$$maF1 = \frac{1}{C}\sum_{i}^{C} F_1^i \tag{7}$$

**Kappa coefficient (Cohen's Kappa).** The Kappa coefficient measures consistency and serves as an indicator of accuracy. Unlike scoring accuracy, the Kappa coefficient accounts for model bias; specifically, when the sample sizes across categories are unbalanced, the model may disproportionately favor larger categories while neglecting smaller ones. The formula for calculating the Kappa coefficient is as follows:

$$Kappa = \frac{p_0 - p_e}{1 - p_e} \tag{8}$$

Where $p_0$ represents the observed consistency, defined as the proportion of movement scores predicted by the model that match the true scores; this corresponds to the overall classification accuracy. Conversely, $p_e$ denotes the probability of random consistency, which is the expected proportion of movement scores predicted by the model that align with the true scores. The formula for calculating $p_e$ is as follows:

$$p_e = \frac{1}{N}\sum_{i=1}^{N}\left(\frac{a_i}{N} \times \frac{b_i}{N}\right) \tag{9}$$

Where $a_i$ represents the number of true samples in category $i$, $b_i$ denotes the number of predicted samples in category $i$, and $N$ signifies the total sample size.

## Implementation and result analysis

**Experimental setup.** We initially employed the RTMPose to extract human skeleton from FMS image sequences. Subsequently, we calculated the cosine similarity distance for each image against the standard action keyframe, identifying the image with the minimum distance as the keyframe. Following this, we computed the action scoring indicators for the keyframe based on the scoring rules established by experts. Finally, these indicators were integrated into the prompt to provide fine-grained scoring and explanatory feedback for the FMS movement. The overall framework of this study is illustrated in Fig 4.

**Method comparison.** To measure the performance of the framework, we extracted the action scores of the framework output in S1 Code. We report in Table 3 the performance of the proposed framework when evaluating each action. Then in Table 4, the performance of the framework with FMS evaluation methods in other studies and the performance of different evaluation methods in other studies are compared. Overall, our proposed framework outperforms other research methods. Moreover, this study not only provides action ratings, but also provides a fine-grained interpretation of the ratings.

Table 4 presents a performance comparison between the automated FMS evaluation method utilizing LLMs and other FMS evaluation methods based on machine learning and deep learning from extant studies, focusing on coarse-grained scoring. Our framework achieves an accuracy of 0.91, which is the highest among all compared methods, signifying a significant advantage in accurately scoring FMS actions. The maF1 score and Kappa coefficient of our framework are 0.87 and 0.82, respectively, slightly lower than those of the methods by Lin et al. [11] and the Dual-Stream Network [13], suggesting that our method may be more effective for imbalanced data or multi-class scoring tasks. The findings indicate that, in

Table 3. Assessment performance of the framework on each movement.

| Movement | Accuracy | maF1 | Kappa |
|---|---|---|---|
| m01 | 1.00 | 1.00 | 1.00 |
| m02 | 0.99 | 0.92 | 0.85 |
| m03 | 0.85 | 0.75 | 0.62 |
| m04 | 0.83 | 0.82 | 0.69 |
| m05 | 0.92 | 0.8 | 0.85 |
| m06 | 0.79 | 0.72 | 0.55 |
| m07 | 0.87 | 0.8 | 0.74 |
| m08 | 0.89 | 0.85 | 0.77 |
| m09 | 0.87 | 0.82 | 0.79 |
| m10 | 0.87 | 0.83 | 0.80 |
| m11 | 0.73 | 0.7 | 0.58 |
| m12 | 1.00 | 1.00 | 1.00 |
| m13 | 1.00 | 1.00 | 1.00 |
| m14 | 1.00 | 1.00 | 1.00 |
| m15 | 1.00 | 1.00 | 1.00 |
| Avg. | 0.91 | 0.87 | 0.82 |

Table 4. Comparison of the framework with existing FMS assessment methods.

| Method | Accuracy | maF1 | Kappa |
|---|---|---|---|
| Improved GMM [10] | 0.80 | 0.77 | 0.67 |
| Xiuchun Lin [11] | 0.88 | 0.88 | 0.82 |
| GCN-based [12] | – | 0.70 | 0.54 |
| Dual-Stream Network [13] | 0.89 | 0.89 | 0.83 |
| Ours | 0.91 | 0.87 | 0.82 |

comparison to deep learning-based action quality assessment methods, the LLMs-based action quality assessment method, which integrates domain knowledge and expert rules, demonstrates comparable performance.

## Discussion

### Dataset

The granularity of a dataset, that is, the detail in data annotation, significantly impacts the research and application within the domain of AQA. Fine-grained datasets have significant benefits in AQA, as they provide more comprehensive annotation details, thereby enhancing the precision of evaluations and the interpretability of the models. Constrained by the level of granularity in existing datasets, the majority of current investigations into AQA remain focused on the estimation of action scores or action levels [23], and few studies carry out fine-grained AQA.

Coarse-grained datasets are relatively straightforward to collect and label. Currently, coarse-grained datasets can be categorized into three distinct groups based on the ground truth: standard action similarity evaluation, grade evaluation and regression-based score evaluation. Within the dataset presented in Table 2, the ground truth for Fitness-28 [18] is the canonical action samples of professional coaches. This dataset is comprised of 28 types

of fitness actions, with a sample size totaling 7530 action videos. A depth camera is utilized to acquire depth data of actions. Employing such datasets, actions are assessed in a coarsely-grained manner through action similarity evaluation. The ground truth for UI-PRMD [21] is the grade evaluation, this dataset contains 10 healthy subjects repeatedly execute 10 distinct physical therapy actions 10 times, accumulating to 1000 action samples. These action samples are bifurcated into two classifications: standard actions and non-standard actions. The ground truth of 3D-Yoga [22] is the regression score, this dataset contains 117 types of yoga actions, with over 3792 action samples and 16668 key frames in total. This dataset provides hierarchical category labels and quality score labels for each action sample, which are adjudicated by three experienced yoga instructors, adhering to standardized yoga pose difficulty coefficients and completion scoring criteria developed by the instructor.

LLM-FMS can be categorized within the category of graded evaluation. In accordance with the FMS scoring guidelines, experts assessed the quality of action execution within keyframes ranging from 1 to 3, encompassing a total of three distinct levels; the higher the score, the higher the quality is deemed to be of the action completion within the subjects' keyframes. Furthermore, experts were engaged to provide detailed annotations of action scoring features in adherence to FMS action evaluation protocols, as well as annotations indicating the body parts associated with the scoring features. Leveraging these two types of detailed semantic annotations, this study conducted an automated, detailed FMS evaluation with the aid of the contextual semantic reasoning capabilities of LLMs, addressing the limitation that, previously, only action scores could be assigned in prior research endeavors.

Constructing fine-grained datasets is an intricate process encompassing numerous challenges associated with data acquisition, processing, labeling, and ensuring data quality. The initial phase in building a dataset entails gathering action data. Currently, there are principally two avenues for acquiring raw data. One involves harvesting data from online media platforms, which is common in the field of sports competitions, such as FineGym [24] and FineDiving [23]. The alternative involves enlisting subjects to execute actions as required, which is common in sports fitness, physical rehabilitation and similar fields. Our dataset is classified within this latter category. This approach is not without its challenges, including the substantial expense associated with participant recruitment, the presence of participants who may lack the necessary qualifications, the complexity of managing the experimental milieu, and the stringent requirements of ethical scrutiny. To oversee the integrity of data acquisition, we engaged FMS specialists to preselect eligible candidates and to establish a dedicated testing site equipped with apparatus specifically for data gathering purposes. The second step is how to define fine-grained semantic labels. Our study proposes a two-tiered fine-grained semantic labels with reference to FMS evaluation rules. The third step is to annotate the data. The process of annotating data for AQA mandates individuals equipped with an adequate level of expertise, rather than simply relying on crowdsourcing platforms. We engaged FMS experts to conduct the annotation, thereby ensuring the stringent nature of the annotation process. Finally, we employ double checking to ensure the quality of annotations, since fine-grained annotations are more error-prone.

## Comparison of methods

Deep learning-based methods, despite their powerful feature extraction and end-to-end learning capabilities, can automatically learn and extract complex action features from data without the need for manual feature design. However, the intricate and opaque internal decision-making mechanisms of these methods hinder their interpretability. The methodologies presented in [11] and [13] utilize I3D networks for the extraction of action features from videos, facilitating subsequent feature learning and scoring processes. The MVDNN model

introduced by [12] integrates deep learning features with manually crafted features to extract FMS action characteristics from multi-view and multi-modal action skeleton data.

Conversely, our framework, which leverages domain knowledge and the contextual learning and logical reasoning capabilities of LLMs, demonstrates robust performance in the coarse-grained automated FMS scoring task. Furthermore, our framework provides fine-grained semantic interpretation, enabling the reasoning of scoring nuances and the identification of body parts that significantly contribute to the assessment, in accordance with established rules. It is important to note that this paper's analysis is based on keyframe image datasets, unlike other studies that have employed video datasets, potentially leading to discrepancies in comparative results.

### Limitations

To align with the logical reasoning capabilities of existing LLMs, the dataset proposed in this paper consists of action keyframe images. Future research can leverage the video processing capabilities of LLMs for fine-grained assessments based on FMS videos. It is important to note that fine-grained annotations necessitate manual decomposition and professional marking. Another limitation is the prompt designs relies on expert experience, and different prompting strategies and LLMs can lead to significant performance differences.

### Future work

In the future, we will leverage the contextual learning ability and logical reasoning ability of LLMs to realize the fine-grained action quality assessment research of more diverse sports. On the other hand, the action evaluation research based on LLMs multi-modal fusion can be explored with the help of LLMs' ability to understand the action behavior in images and videos.

## Conclusion

In this paper, we propose the first fine-grained FMS action keyframe dataset, LLM-FMS, designed for the assessment of FMS movement quality using LLMs. To enhance the interpretability of the assessment framework, the dataset includes scoring detail annotation labels and body part information labels. Furthermore, we propose a fine-grained FMS quality assessment framework based on LLMs, leveraging their logical reasoning capabilities to improve the semantic interpretability of movement quality assessments. This approach renders the reasoning process more transparent and achieves significant advancements over existing AQA methods, moving beyond simple movement scoring or categorical grading.

### Publicly accessible data

This work presents a fine-grained FMS dataset, which can be accessed via the figshare repository: https://doi.org/10.6084/m9.figshare.c.7601630.v1.

## Supporting information

**S1 Table. Demographic information of the subjects.** Age, sex, height, weight, and BMI of the subjects.
(XLSX)

**S1 Text. Threshold conditions for angle and distance of FMS action keyframes.**
(PDF)

**S2 Text. Detailed rules for FMS movements.**
(PDF)

**S3 Text. LLMs' Prompt for FMS movements.**
(PDF)

**S1 Code. Framework performance evaluation scripts.**
(PY)

## Acknowledgements

The authors extend their gratitude to the numerous colleagues, students, and library and faculty staff who contributed to the FMS action sample collection in the service of this project. Additionally, appreciation is expressed to the Intelligent Sports Engineering Laboratory for furnishing the experimental facilities essential for this study. In particular, the authors extend their gratitude to Dr. Xuemei Li, Dr. Dapeng Bao, and Mr. Peng Zhang for their substantial contributions to the annotation of the dataset.

## Author contributions

**Conceptualization:** Qingjun Xing, Ping Guo, Yan-Fei Shen.

**Data curation:** Xu-Yang Xing, Ping Guo.

**Formal analysis:** Qingjun Xing, Xu-Yang Xing.

**Funding acquisition:** Yan-Fei Shen.

**Investigation:** Qingjun Xing, Zhen-Hui Tang.

**Methodology:** Xu-Yang Xing, Ping Guo.

**Project administration:** Ping Guo.

**Visualization:** Xu-Yang Xing.

**Writing – original draft:** Qingjun Xing.

**Writing – review & editing:** Ping Guo, Yan-Fei Shen.

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
