## [Decision Letter · Decision Letter 0]

27 Nov 2024

PONE-D-24-49253LLM-FMS: A Fine-grained Dataset for Functional Movement Screen Action Quality AssessmentPLOS ONE

Dear Dr. xing,

Thank you for submitting your manuscript to PLOS ONE. After careful consideration, we feel that it has merit but does not fully meet PLOS ONE’s publication criteria as it currently stands. Therefore, we invite you to submit a revised version of the manuscript that addresses the points raised during the review process.

We look forward to receiving your revised manuscript.

Kind regards,

Ananth JP

Academic Editor

PLOS ONE

Journal Requirements:

2. Thank you for stating the following financial disclosure: [This study has been supported by the National Natural Science Foundation of China, Grant No. 72071018, the Fundamental Research Funds for the Central Universities of China, No. 2023GCZX003 (Research on the Nonlinear Accurate Measurement System of Exercise Loads)]. Please state what role the funders took in the study. If the funders had no role, please state: "The funders had no role in study design, data collection and analysis, decision to publish, or preparation of the manuscript." If this statement is not correct you must amend it as needed. Please include this amended Role of Funder statement in your cover letter; we will change the online submission form on your behalf

3. Thank you for stating the following in the Acknowledgments Section of your manuscript: [This study has been supported by the National Natural Science Foundation of China, Grant No. 72071018, the Fundamental Research Funds for the Central Universities of China, No. 2023GCZX003 (Research on the Nonlinear Accurate Measurement System of Exercise Loads).] We note that you have provided funding information that is not currently declared in your Funding Statement. However, funding information should not appear in the Acknowledgments section or other areas of your manuscript. We will only publish funding information present in the Funding Statement section of the online submission form. Please remove any funding-related text from the manuscript and let us know how you would like to update your Funding Statement. Currently, your Funding Statement reads as follows: This study has been supported by the National Natural Science Foundation of China, Grant No. 72071018, the Fundamental Research Funds for the Central Universities of China, No. 2023GCZX003 (Research on the Nonlinear Accurate Measurement System of Exercise Loads)] Please include your amended statements within your cover letter; we will change the online submission form on your behalf.

4. Thank you for uploading your study's underlying data set. Unfortunately, the repository you have noted in your Data Availability statement does not qualify as an acceptable data repository according to PLOS's standards. At this time, please upload the minimal data set necessary to replicate your study's findings to a stable, public repository (such as figshare or Dryad) and provide us with the relevant URLs, DOIs, or accession numbers that may be used to access these data. For a list of recommended repositories and additional information on PLOS standards for data deposition, please see https://journals.plos.org/plosone/s/recommended-repositories .

5. Please include captions for your Supporting Information files at the end of your manuscript, and update any in-text citations to match accordingly. Please see our Supporting Information guidelines for more information: http://journals.plos.org/plosone/s/supporting-information .

Reviewers' comments:

Reviewer's Responses to Questions

**Comments to the Author**

1. Is the manuscript technically sound, and do the data support the conclusions?

Reviewer #1: Yes

Reviewer #2: Yes

2. Has the statistical analysis been performed appropriately and rigorously? 

Reviewer #1: Yes

Reviewer #2: Yes

3. Have the authors made all data underlying the findings in their manuscript fully available?

Reviewer #1: Yes

Reviewer #2: Yes

4. Is the manuscript presented in an intelligible fashion and written in standard English?

Reviewer #1: Yes

Reviewer #2: Yes

5. Review Comments to the Author

Reviewer #1: Author built a dataset for Functional Movement Screen with the help of FMS video to analyze the movement patterns. Author uses computer technology and artificial intelligence to automate and enhance functional movement screening for improving assessment efficiency, minimizing subjective bias, and broadening service coverage. This paper also proposes a framework that leverages the reasoning capabilities of LLMs to evaluate action quality on the LLM-FMS dataset. But, this work has few drawbacks that needs to be resolved.

1. Abstract should higlight the probem statement and address the experimental results obtained.

2. The structure of the manuscript needs to be mentioned.

3. Literature review should discuss the several work related to this field.

4. Experiments should be done by comparing with several works.

5. Result analysis needs to be explained elaborately.

6. Future work needs to be discussed.

Reviewer #2: The abstract clearly outlines the paper's purpose, which is to introduce a detailed dataset, LLM-FMS, designed for evaluating the quality of Functional Movement Screen (FMS) actions. Still, the abstract might improve by including more details regarding the dataset's scale (e.g., sample size, labeled features) and the precise methodologies employed for quality assessment, as this would enhance the context of its contributions.

The introduction effectively establishes the importance of FMS in fields such as sports science and healthcare. It emphasizes the necessity for automated assessment through machine learning. The justification for concentrating on FMS, compared to more extensive human action assessment datasets, deserves more clarification.

The paper cites previous research in action quality assessment and generic human movement datasets. However, a more thorough examination of pertinent literature, including relevant datasets and specific models for movement analysis, would add strength for the necessity of LLM-FMS. Few more recent literatures can be reviewed and a summary of the literature survey along with the inference can be given at the end of the survey.

The dataset description is given well and also the table compares the existing rehabilitation and fitness datasets with LLM-FMS. The existing dataset details can be given in the literature survey part too to add weightage.

The results section convincingly demonstrates the effectiveness of LLM-FMS for training action quality models. The discussion does not delve into the challenges faced during data collection and labeling, which are often critical in fine-grained datasets.

The conclusion summarizes the dataset's contributions well but could do more to address potential limitations. Future scope can be included and also it could explore the inclusion of sensor data (e.g., accelerometers) or the expansion of the dataset to include non-standard FMS movements.

Overall strengths of the paper are as follows:

1. The paper tackles a relevant and underexplored problem: fine-grained action quality assessment in FMS.

2. The multi-modal approach to dataset creation enhances its utility across various machine learning techniques.

3. The benchmarking experiments provide a solid foundation for future work.

The overall weakness of the paper are as follows:

1. Limited theoretical grounding for the dataset's novelty and importance in the broader action quality assessment landscape.

2. The lack of comparison with existing datasets and methods weakens the paper's positioning.

Few recommendations:

1. Provide a more detailed literature review with explicit comparisons to existing datasets and methodologies.

2. Include demographic and diversity statistics for participants in the dataset description.

This research significantly contributes to the domain of action quality assessment with its detailed LLM-FMS dataset. To enhance its effectiveness, the authors should develop the theoretical foundation, clarify dataset particulars, and strengthen the experimental section. These improvements could establish the work as an essential component for research in automated functional movement analysis.

6. PLOS authors have the option to publish the peer review history of their article (what does this mean? ). If published, this will include your full peer review and any attached files.

**Do you want your identity to be public for this peer review?** For information about this choice, including consent withdrawal, please see our Privacy Policy .

Reviewer #1: No

Reviewer #2: **Yes: ** Dr. Oswalt Manoj S

---

## [Author Response · Author response to Decision Letter 1]

14 Jan 2025

Review Comments to the Author

Reviewer #1:

1. Abstract should higlight the probem statement and address the experimental results obtained.

Response: Thank you for your valuable feedback on our abstract. We have carefully revised the abstract to ensure that it clearly highlights the problem statement and addresses the experimental results obtained.

2. The structure of the manuscript needs to be mentioned.

Response: Thank you for your feedback regarding the structure of our manuscript. We have carefully reviewed the structure and have made the necessary adjustments to ensure that it aligns with the requirements of PLOS ONE, as outlined in the provided style templates:

PLOSOne_formatting_sample_main_body.pdf

PLOSOne_formatting_sample_title_authors_affiliations.pdf

We have also ensured that the manuscript is formatted according to PLOS ONE's style requirements, including font sizes, line spacing, and margins. Additionally, we have checked the file naming conventions and have made sure that all supporting information files are named appropriately and are referenced correctly within the manuscript.

3. Literature review should discuss the several work related to this field.

Response: Thank you for your suggestions on strengthening the literature review section of our manuscript. We have significantly revised the Introduction, focusing in particular on the fourth and sixth paragraphs of the Introduction, to provide a more comprehensive discussion of related work in the field of intelligent evaluation of FMS and the application of LLMs in motion analysis, prediction, and control.

In the fourth paragraph of the introduction, we extend the discussion of intelligent evaluation of functional motor screens (FMS) using deep learning methods. We also added a new study from 2024 that explores the use of deep learning techniques for FMS evaluation [line number: 85-90]. The following new articles were added:

[1] Lin X, Liu Y, Feng C, et al. Automatic Evaluation Method for Functional Movement Screening Based on Multi-Scale Lightweight 3D Convolution and an Encoder–Decoder[J]. Electronics, 2024, 13(10): 1813.

Furthermore, in paragraph VI, we present recent advances in the use of large language models (LLMs) for motion analysis, prediction, and control. In this paper, we discuss in detail a recent study of LLMs for action prediction in fine-grained videos, which demonstrated the effectiveness of LLMs for understanding human motion and videos, highlighting significant performance improvements over prior approaches [line: 99-113]. This section now presents more clearly the state of the art of research in this area and the potential of LLMS. The following new articles were added:

[1] Zhao Q, Wang S, Zhang C, et al. Antgpt: Can large language models help long-term action anticipation from videos?[J]. arXiv preprint arXiv:2307.16368, 2023.

[2] Joublin F, Ceravola A, Smirnov P, et al. CoPAL: corrective planning of robot actions with large language models[C]//2024 IEEE International Conference on Robotics and Automation (ICRA). IEEE, 2024: 8664-8670.

4. Experiments should be done by comparing with several works.

Response: Thank you for your suggestion to compare our experimental results with several related works. We have addressed this by comparing the FMS coarse-grained action scoring results obtained using our LLM-based assessment framework with the results from several recent FMS action assessment studies. The comparison results are presented in Table 4, where we have evaluated the performance using three key metrics: Accuracy, macro F1-score (maF1), and kappa coefficient.

5. Result analysis needs to be explained elaborately.

Response: Thank you for your valuable feedback regarding the need for a more elaborate explanation of our result analysis. We reinterpret the results of the study, explaining the evaluation effect of the method on each action of FMS, and the performance comparison between the method of this study and other studies on the evaluation method of FMS action rating. [line: 352-371].

6. Future work needs to be discussed.

Response: Thank you for your suggestion to include a discussion on future work in our manuscript. We have addressed this by adding a dedicated section in the Discussion part of our paper, starting at line [454-458], where we outline our plans for future research directions. In the future, we will leverage the contextual learning ability and logical reasoning ability of LLMs to realize the fine-grained action quality assessment research of more diverse sports. On the other hand, the action evaluation research based on LLMs multi-modal fusion can be explored with the help of LLMs' ability to understand the action behavior in images and videos.

Reviewer #2:

The overall weakness of the paper are as follows:

1. Limited theoretical grounding for the dataset's novelty and importance in the broader action quality assessment landscape.

Response: Thank you for your suggestion. In the first four paragraphs of the Introduction [line: 37-98], we discuss the theoretical basis of the dataset in the field of motion quality assessment, highlighting the importance of this dataset in the field of sports rehabilitation. In addition, we discuss the novelty of this dataset in the Discussion section [line: 374-408].

2. The lack of comparison with existing datasets and methods weakens the paper's positioning.

Response: Thank you for your suggestion. We add a comparison and discussion with existing action quality assessment datasets in the Discussion section [line: 374-429] and a comparison with existing automated FMS assessment methods in the discussion section [line: 430-446], thus highlighting the novelty of the LLM-FMS dataset and the LLMs-based action assessment framework in this study.

Few recommendations:

1. Provide a more detailed literature review with explicit comparisons to existing datasets and methodologies.

Response: Thank you for your suggestion. We provide a more detailed literature review of research on automated FMS evaluation in the fourth introductory paragraph [line: 66-90], and a literature review of recent research on in-video action prediction and motion control with the aid of LLMs in the sixth paragraph [line: 99-113]; A comparison of existing datasets [line: 374-397] and methods [line: 430-446] is also presented in the discussion section.

2. Include demographic and diversity statistics for participants in the dataset description.

Response: Thank you for your suggestion. In the "Basic information of the dataset" section of the "LLM-FMS fine-grained dataset" section of the Methods section [line 190], the demographic and diversity statistics of the subjects of the dataset are explained. Specific statistical information is attached to S1 Table-Demographic Information of the subjects.

---

## [Editor Report · Decision Letter 1]

17 Jan 2025

LLM-FMS: A Fine-grained Dataset for Functional Movement Screen Action Quality Assessment

PONE-D-24-49253R1

Dear Dr. xing,

We’re pleased to inform you that your manuscript has been judged scientifically suitable for publication and will be formally accepted for publication once it meets all outstanding technical requirements.

Kind regards,

Ananth JP

Academic Editor

PLOS ONE
---

## [Editor Report · Acceptance letter]

PONE-D-24-49253R1

PLOS ONE

Dear Dr. Xing,

I'm pleased to inform you that your manuscript has been deemed suitable for publication in PLOS ONE. Congratulations! Your manuscript is now being handed over to our production team.

Kind regards,

on behalf of

Dr. Ananth JP

Academic Editor

PLOS ONE